# The Necessity of LED to Ambient Light Ratio Optimization for Vehicular Optical Camera Communication

**DOI:** 10.3390/s20010292

**Published:** 2020-01-04

**Authors:** Trong-Hop Do, Myungsik Yoo

**Affiliations:** School of Electronic Engineering, Soongsil University, Seoul 06978, Korea; dotronghop@gmail.com

**Keywords:** optical, visible light, camera, vehicle, communication, LED, ambient light, optimization

## Abstract

In vehicular optical camera communication (VOCC) systems, LED panels are used to transmit visible light signals which are captured by cameras. The logic bits 1 and 0 are represented by the On and Off status of the LEDs in the panel. The bit error rate (BER) of the system is directly proportional to the distinguishability of the On and Off LEDs in the received LED panel images. The signal quality is commonly believed in telecommunications to improve with a higher transmitted power. Therefore, one might expect to get a lower BER in VOCC systems by simply using more powerful LED lights. However, this is not the case with VOCC systems. This paper shows that the LED distinguishability is simultaneously determined by two factors: The LED extinction ratio and LED interference. The former needs to be kept high and the latter kept low for better LED distinguishability. The problem is that both the extinction ratio and interference increase with the ratio of LED light to ambient light (L2A). Consequently, an optimal L2A ratio exists to achieve the optimal balance between the positive impact of the extinction ratio and the negative impact of the interference. This can bring about the lowest BER without changing the system’s data rate. In addition, this paper shows that the optimal L2A ratio varies according to the interval between the LEDs in a panel. We analyze the effect of the L2A ratio and LED interval on LED distinguishability. We then formulate a constrained optimization problem to find the optimal L2A ratios corresponding to different LED intervals. The simulation results verify the necessity of LED to ambient light ratio optimization as it can bring about the lowest BER without scarifying other aspects of the VOCC system.

## 1. Introduction

Vehicular communication is a type of short- to medium-range communication system for exchanging traffic information and safety warnings among vehicles [1,2]. Vehicular communication has many useful applications. A vehicle can, for instance, transmit a signal for lane changing permission and wait for a confirmation sent from other vehicles. When there is a need for emergency braking, accelerating or decelerating, warning signals can be broadcasted to other vehicles through vehicular communication. When used with vehicle tracking, a vehicle’s ID can be sent to other vehicles to build maps of traffic on the street. Coupling vehicular communication with vehicle tracking and other sensing technologies such as vehicle tracking, lane detection, pedestrian detection, etc., will facilitate safer, more coordinated transportation networks, and will ultimately spur autonomous vehicle networks and intelligent transport systems. In the last few years, an emerging technology known as optical camera communication (OCC) has been regarded as a potential candidate for vehicular communication thanks to many advantages [3,4,5].

In vehicular optical camera communication (VOCC), the visible light signal is transmitted using LED panels and is received using dashboard cameras, which are both available in vehicles. The availability of the transmitter and receiver creates a tremendous cost advantage for VOCC. However, the most important advantage that distinguishes VOCC from other vehicle communication technologies is the perfect compatibility with other essential technologies for intelligent transport systems and autonomous vehicle networks. In the near future, it is likely that vision-based technologies such as traffic sign detection, street lane detection, pedestrian detection, and vehicle tracking, can be implemented in every vehicle. These technologies employ the same cameras and image processors required for VOCC [6,7]. In particular, VOCC can work in great harmony with vehicle tracking, which is one of the most crucial components in a self-driving vehicle network. This is due to key procedures in vehicle tracking, namely vehicle image coordinate detection and vehicle identification, which are already achieved through VOCC. As a result, both hardware and software implementation costs for whole-vehicle communication and tracking systems can be substantially lowered by using VOCC for vehicle communication over other technologies [8].

A common understanding in telecommunications is that increasing the power of a transmitter would result in better signal quality. Therefore, one might expect to get a lower bit error rate (BER) in VOCC by simply using more powerful LED lights; however, this is not the case. In a VOCC system, the logic bits 0 and 1 are conveyed by the Off and On status of LEDs in the panel. The BER of the system is determined by the distinguishability of the On and Off LED statuses in the received images. This paper shows that the LED distinguishability is mutually determined by two factors: The LED extinction ratio and LED interference. Note that “extinction ratio’’ is a term used in optical communication to express the ratio between two received optical powers that represent the logic levels 1 and 0. In VOCC, the LED extinction ratio is the ratio between the greyscales of the On and Off LEDs in the received LED panel image. LED interference is the interference between adjacent LEDs in the received image. Both factors are governed by the LED to ambient light (L2A) ratio. More specifically, due to the nature of image acquisition, both the extinction ratio and interference increase as the L2A ratio increases. While the increase in the LED extinction ratio reduces the BER, the increase in LED interference escalates it. Consequently, as the L2A ratio increases, the BER initially drops until the L2A ratio reaches an optimal level. When the L2A ratio exceeds this optimal level, the BER escalates because the positive impact of the increasing extinction ratio is outweighed by the negative impact of the increasing interference. Ambient light is uncontrollable in outdoor environments where VOCC systems operate. However, the light output from LEDs can be adjusted to the optimal L2A ratio to achieve the lowest BER at the same data rate. This paper shows that the optimal level of the L2A ratio is not fixed, but varies according to the interval between LEDs in the panel.

The contribution of this paper is multifold. Firstly, to the best of our knowledge, this paper is the first work in literature to discuss the effect of the L2A ratio on BER in VOCC systems. Through theoretical analysis, this paper elucidates the L2A ratio and LED interval as the two primary parameters governing the BER of a system. This analysis also reveals that the lowest BER can be achieved through maximizing the ratio of the between-class variance to the within-class variance (B2W) of the On and Off LED features in received images. Based on that, we formulate a constrained optimization problem to finding the optimal L2A ratios in order to maximize the B2W variance ratio subject to different LED intervals. The optimization result, which is the optimal ranges of L2A ratios at different LED intervals, is achieved through simulation. In practice, the L2A ratio optimization process is performed and the optimal values of L2A ratios are obtained at the hands of the manufacturers in the offline phase before the VOCC components are released to the market. This process can be accomplished using the method of maximizing the B2W variance ratio as proposed in the paper. In the online phase when the VOCC system is operating, the luminance of the LEDs can be adjusted to the optimal level corresponding to the current ambient light and LED intervals. Note that the level of ambient light can be estimated easily using built-in light metering systems in commercial cameras. In this paper, the simulation result shows that by using the optimal L2A ratio, a much lower BER of the VOCC system can be achieved without having to change the data rate.

## 2. Fundamentals of VOCC System Architecture

### 2.1. System Architecture

The VOCC system architecture is described in Figure 1. Firstly, the transmitted bits are applied to error coding and modulation. Then, the bits are conveyed by the statuses On and Off of LEDs in the LED panel. Afterwards, the image of the LED panel is captured using a camera in other vehicles. To obtain the transmitted bits, the captured images are processed using LED panel detection and LED bit detection algorithms. After applying error decoding, the transmitted bits are received.

In LED panel detection, the positions of the LED panels in the image are located. In LED bit detection, the detected LED panels are closely examined to determine the On or Off status of each LED chip in the panels. These statuses are then translated to the logical bits 1 and 0.

### 2.2. LED Bit Detection in VOCC

In VOCC systems, transmitted data are obtained through the LED bit detection step, as shown in Figure 2. Upon locating the position of the entire LED panel in the image, the LED panel is divided into equal sub-regions. It is assumed that the number of LEDs in the panel is known beforehand. In Figure 2, the LED panel is divided into 8 × 8 equal sub-regions because there are 8 × 8 LEDs in the panel. Each of these sub-regions contains the information of an LED in the panel. From each sub-region, two LED features, namely the average greyscale ratio (AGR) and the gradient radial inwardness (GRI), can be obtained [9]. From these two features, algorithms such as Linear Fisher Discriminant Analysis or K-mean clustering [10] can be used to distinguish the status On and Off of each LED.

The AGR is calculated as follows:(1)$$AGR=\bar{G_{led}}/\bar{G_{panel}},$$
where $\bar{G_{led}}$ is the average greyscale of an LED and $\bar{G_{panel}}$ is the average greyscale of the entire LED panel.

The GRI is calculated as follows:(2)$$GRI=\frac{\sum \nabla f(x,y)\cdot r(x,y)}{number\;of\;pixels},$$
where ∇f(x,y) denotes the gradient vector at each pixel inside the LED region, (·) denotes the dot product operation, and r(x,y) denotes the normalized radial inward vector at the corresponding pixel.

The gradient vector is given as:(3)$$\nabla f\left(x,y\right)=\left[\begin{array}{c} f\left(x+1\right)-f\left(x-1\right)\\ f\left(y+1\right)-f\left(y-1\right) \end{array}\right],$$
where f(x,y) is the greyscale at the coordinate (x,y) in the image.

The normalized radial inward vector is given as:(4)$$r\left(x,y\right)=\frac{\left(x_{c}-x,y_{c}-y\right)}{∥\left(x_{c}-x,y_{c}-y\right)∥}$$
where $\left(x_{c},y_{c}\right)$ are the coordinates of the center pixel of the LED and ∥v∥ denotes the magnitude of the vector v.

## 3. LED to Ambient Light Ratio Optimization

### 3.1. The Necessity of LED Dimming and the Proper Exposure Mechanism of Camera

The two most important differences between the VOCC system in this paper compared to that in existing works are the dimming capability of LED and the exposure setting of cameras. Regarding the LED dimming capability, this paper assumes that LEDs are dimmable. In existing papers related to VOCC, the dimming aspect of LED has not been concerned. However, in practice, for the sake of driving safety, LEDs must not be much brighter than ambient light. The iris control in human eyes works in the principle of closing in bright ambient light to allow less light through the pupil and opening in dim ambient light to allow more light to enter the pupil. Because of this reason, vehicle headlights which appear dim in daytime are still much brighter than the ambient light in nighttime and thus can cause serious flare for humans. Similarly, daytime running lights, which are very suitable for using as transmitters in VOCC systems thanks to their brightness that make them clearly visible even in daytime must not be used at night. Consequently, in the VOCC system, the luminance of LED must be changed corresponding the ambient light to maintain the ratio between their brightness within a safe range.

Regarding the exposure setting of cameras, this aspect has also not been concerned in existing papers on VOCC. The experiment and simulation in those papers considered only the communication between two vehicles. Therefore, the exposure of the camera was simply adjusted corresponding to the LED luminance. This mechanism helps prevent the LED blooming caused by bright LEDs to the captured images. However, this mechanism is not practical. The reason is because the vehicle communicates with multiple other vehicles, of which the LEDs might have different levels of luminance. Moreover, in practice, while the exposure of the camera must be set before capturing the image, the luminance of LEDs can only be estimated after capturing the image. Furthermore, in our vision for intelligent transport system, the video output from the camera should be used for not only communication but also detection of lanes, pedestrians, and traffic signs, and other kinds of object detection required for autonomous vehicles. Changing the camera exposure to the LED luminance would not ensure sufficient greyscale levels for all objects in the image to be detectable. Consequently, in this paper, the camera exposure is assumed to adjust automatically to ambient light.

As the ambient light is the common factor between vehicles, the luminance of LEDs and the exposure of cameras in all vehicles in the road can all be adjusted to the current ambient light to ensure the traffic safety and the ease for camera exposure setting in all vehicles. These mechanisms also bring about the harmony between the VOCC system and other object detection components in autonomous vehicle as the output images from the camera can be used for both communication and object detection. It is the mechanisms of LED dimming and camera auto exposure adjusted to the ambient light that lead to the optimization problem concerned in this paper.

### 3.2. A Brief Introduction to the Optimization Problem

The performance of a VOCC system is evaluated through two aspects: Data rate and BER. Regarding the former, a higher data rate without scarifying the BER of the VOCC system can be achieved by using cameras with higher frame rates or larger LED panels to accommodate more LED chips. Cameras with higher frame rates and more powerful processors are obviously costly, and vehicles might not have enough space to install a large LED panel. Consequently, increasing the data rate in the VOCC system usually involves the problem of balancing between the required data rate and the implementation costs of particular systems; this is not the concern of this paper. The second aspect—the BER—on the other hand, can be improved by optimizing the system parameters without having to increase the implementation costs. Therefore, this paper is concerned with optimizing VOCC system parameters to obtain the optimal BER at the same implementation costs.

Among the many parameters, L2A ratio and LED spacing are the ones with decisive effects on BER. As mentioned earlier, the L2A ratio is the ratio between LED and ambient light. LED spacing is the amount of space between LEDs in the captured LED panel image. Figure 3 shows a top-down explanation on the deciding effects of the L2A ratio and LED spacing on the LED bit detection accuracy. More specifically, the LED bit detection accuracy is determined by the characteristics of LED features extracted from the received image. These characteristics include between-class variance (Bvar) and within-class variance (Wvar) of the On and Off LED features. These variances are in turn determined by the characteristics of On and Off LEDs in the received image. These imaging characteristics of LED includes LED extinction ratio and LED interference. The characteristics of LEDs in the received image are in turn determined by system parameters including the L2A ratio and LED spacing. The following subsections provide details of the relationships shown in Figure 3.

### 3.3. A Top-Down Explanation on How L2A Ratio and LED Density Determine Bit Detection Accuracy

#### 3.3.1. Effect of Bvar and Wvar on LED Bit Detection Accuracy

In all problem of data classification, the data would be classified better when the data within each class is close to each other while the data from different class is far to each other. Between-class variance (Bvar) and within-class variance (Wvar) are the statistical quantification for these characteristics of data. As illustrated in Figure 4, Bvar indicates the separation in the features of two classes while Wvar indicates the scattering of features within each class. LED bit detection is indeed a problem of classifying On and Off LED from obtained LED features. Consequently, the bit detection accuracy is high when Bvar is large and Wvar is small.

Let $x=(x_{AGR},x_{GRI})$ denote the features of an LED, $x_{On}$ and $x_{Off}$ denote the On and Off states of the LED features. The between-class variance $\sigma_{b}$ is calculated as:(5)$$\sigma_{b}={\left(\mu_{On}-\mu_{Off}\right)}^{2},$$
where $\mu_{On}=E\left[x_{On}\right]$ and $\mu_{Off}=E\left[x_{Off}\right]$ are the means of the On and Off LED features.

The within-class variance $\sigma_{w}$ is calculated as:(6)$$\sigma_{w}=E\left[{\left(x_{On}-\mu_{On}\right)}^{2}\right]+E\left[{\left(x_{On}-\mu_{On}\right)}^{2}\right].$$

To achieve the best distinguishability between On and Off LEDs, the variance ratio of between-class to within-class variances needs to be maximized:(7)$$R_{B2W}=\frac{\sigma_{b}}{\sigma_{w}},$$

#### 3.3.2. Effect of Extinction Ratio and Interference on Bvar and Wvar

Bvar and Wvar of the On and Off LED features are determined by the characteristics of LEDs in the received image. In VOCC, these imaging LED characteristic includes LED extinction ratio and LED interference as illustrated in Figure 5.

Obviously, the Bvar is larger when the extracted features of On LEDs are more different compared to that of Off LEDs. After all, the On LEDs are differentiated to Off LEDs in the image only because of the difference in their greyscale. Therefore, the more difference between On and Off LED greyscale, the larger Bvar is.

In the filed of image processing, the greyscale difference between On and Off LEDs can be expressed by the term contrast. However, in optical communication, the difference between two received optical powers that represent the logic levels 1 and 0 is usually expressed by the term extinction ratio. In VOCC, the logic level 1 is conveyed by the On LEDs while the logic level 0 is conveyed by the Off LEDs in the captured image. Therefore, in the VOCC system, the difference in greyscales of the On and Off LEDs in the received LED panel image is expressed by the term extinction ratio:(8)$$r_{e}=\frac{G_{On}}{G_{Off}},$$
where $r_{e}$ is the LED extinction ratio, $G_{On}$ and $G_{Off}$ are the greyscales of the On and Off LEDs, respectively.

As shown in Figure 5, a higher extinction ratio creates a significant difference in the greyscale of On and Off LEDs in the image, and thus a larger Bvar, between the On and Off LED features.

Besides LED extinction ratio, LED interference is another important characteristic of LEDs in the captured image that has direct influence on the characteristic of extracted LED features. LED interference is defined as the interference between adjacent LEDs in the captured image. Because of the interference, the boundary of On and Off LEDs are not well-defined as illustrated in Figure 6. Consequently, the larger LED interference results in the larger variance of the extracted features of On and Off LEDs. As Wvar indicates the homogeneity of the On and Off features, a higher interference creates a larger Wvar.

The impacts of LED extinction ratio and LED interference on Bvar and Wvar can be described in Figure 6.

#### 3.3.3. Effect of L2A Ratio and LED Spacing on Extinction Ratio and Interference

##### (a) Effect of L2A Ratio on Extinction Ratio

As mentioned earlier, LED extinction ratio is the difference in terms of the ratio between the greyscale of On and Off LEDs in the captured image. Intuitively, the greyscale of On and Off LEDs are different only because the intensity of the light coming from On and Off LEDs to the camera are different. In the VOCC system, the difference in the intensity of the light coming from On and Off LEDs can be expressed through the LED to ambient light (L2A) ratio defined as:(9)$$r_{L2A}=\frac{L_{v}}{E_{v}R},$$
where $L_{v}$ is the luminance of an LED, $E_{v}$ is the ambient light illuminance, and *R* is the reflectance of the LED surface.

The direct effect of the L2A ratio on the LED extinction ratio is mathematically described as:(10)$$r_{e}={\left(1+r_{L2A}\right)}^{1/\gamma },$$
where $r_{e}$ is the LED extinction ratio, $r_{L2A}$ is the L2A ratio, and γ is the gamma value used for gamma encoding.

In digital image acquisition, the greyscale value *G* at a pixel in the image sensor at a given ISO speed *S* set by the camera is determined as [11]:(11)$$G=118\times {\left(\frac{H}{H_{SOS}}\right)}^{1/\gamma },$$
where *H* denotes the luminous exposure, $H_{SOS}=10/S$ denotes the indicated luminous exposure at the ISO speed *S* to obtain the greyscale value of 118, and γ denotes the gamma value used for gamma encoding.

The light from an Off LED is the reflected ambient light from the LED surface. Thus, the luminous exposure of an Off LED pixel is calculated as [11]:(12)$$H_{Off}=\frac{10E_{v}Rt}{K\pi N^{2}},$$
where $E_{v}$ denotes the ambient light illuminance, *R* denotes the LED surface reflectance, *t* denotes the camera exposure time, *N* denotes the lens f-number, and *K* denotes the reflected-light meter calibration constant given by camera manufactures.

The light from an On LED is composed of the light emitted by the LED itself and the reflected ambient light from the LED surface. Thus, the luminous exposure of an On LED pixel is determined as [11]:(13)$$H_{On}=\frac{10E_{v}Rt}{K\pi N^{2}}+\frac{10L_{v}t}{KN^{2}},$$
where $L_{v}$ is the luminance of the LED.

Equation (Equation 10) can be derived from Equations (Equation 8), (Equation 9) and (Equation 11)–(Equation 13).

##### (b) Effect of L2A Ratio and LED Spacing on Interference

Interference is caused by image blurriness, which is the result of imperfections in the image acquisition process. These imperfections, including air diffraction and optical aberrations, cause the light to be spread out over a region rather than focused to a point, hence creating image blurriness. As illustrated in Figure 7, these effects can be mathematically described as convolving the incoming image with a Gaussian blur function:(14)$$I^{\prime }\left(r,c\right)=\sum_{i}\sum_{j}I(i,j)\times G(r-i,c-j),$$
where $I^{\prime }$ is the result image, *I* is the incoming image, and (r,c) are the row and column coordinates in the image. *G* is the Gaussian blur function given as:(15)$$G\left(r,c\right)=\frac{1}{2\pi \sigma^{2}}e^{-\frac{r^{2}+c^{2}}{2\sigma^{2}}},$$
where σ is the standard deviation of the Gaussian distribution.

As illustrated in Figure 5 and Figure 7, interference increases the pixel values of Off LEDs around On LEDs. From Equations (Equation 14) and (Equation 15), the interference is directly proportional to the greyscale *I* of On LEDs and the value of the Gaussian function, which is in turn directly proportional to the standard deviation σ and the LED spacing $r^{2}+c^{2}$ between the LEDs in the image. The greyscale *I* of On LEDs is in turn directly proportional to the L2A ratio. Consequently, the interference is determined by the L2A ratio and the LED spacing.

##### (c) Effect of LED Interval and Communication Distance on LED Spacing

As mentioned earlier, LED spacing is the amount of space between LEDs in the images. From the camera geometry, LED spacing is determined as:(16)$$spacing=\frac{intv_{LED}\times f}{d},$$
where $intv_{LED}$ is the LED interval, which is defined as the inter-distance between LED chips in the LED panel, *f* is the lens focal length, and *d* is the communication distance.

From Equation (Equation 16), it can be seen that LED spacing is directly proportional to the LED interval $intv_{LED}$ and lens focal length *f* increase while inversely proportional to the communication distance *d*.

### 3.4. Constraint Optimization Problem of VOCC System

#### 3.4.1. The Idea behind VOCC System Optimization

As explained above, the LED spacing and the L2A ratio are the parameters with decisive influence on the LED bit detection accuracy of VOCC systems. The former has a great impact on the LED bit detection. As the LED spacing increases, the interference between the On and Off LEDs reduces, the B2W variance ratio increases, and thus the LED bit detection accuracy increases. Therefore, one can reduce the BER by increasing the LED spacing. From Equation (Equation 16), this can be achieved by reducing the communication distance or increasing the lens focal length and the LED interval. It is obvious that communication distance is not the parameter to be changed as the vehicles communicate at any distance. The lens focal length is also not a parameter that can be increased freely since it will narrow down the field of view of the camera. LED interval is the parameter that has the best chance to be increased to increase the LED spacing. However, as the physical area of a LED panel in the vehicle is limited, the number of LEDs in the panel must be reduced to increase the LED interval. This leads to a decrease in data rate. Consequently, LED interval should not be considered as variable to be changed to increase the LED bit detection accuracy but a soft constraint with varying values given from each VOCC system.

Even though the L2A ratio has less impact on LED bit detection compared to LED interval, the change of this parameter does not affect the data rate of the VOCC system. Therefore, the L2A ratio is the only variable chosen in this paper for the optimization of the VOCC system.

As a optimization variable, the value of the L2A ratio needs to be optimized to achieve the highest LED bit detection accuracy. The interesting thing about the L2A ratio, which is also the thing that make the L2A ratio optimization worth researching, is its contradicting effects on LED bit detection as shown in Figure 8.

Figure 8 shows the captured images of two LEDs at various conditions of ambient light. In all images, the left LED is Off and the right LED is On. The luminance of the On LED remains the same in all images. Since the illuminiance of the ambient light decreases from the left to the right images, the L2A ratio increases accordingly. Under the direct strong sunlight, the L2A ratio is low and thus both the extinction ratio and interference are low. As the L2A ratio increases, both the extinction ratio and interference increase. Note that in Figure 8e, the illuminance in twilight condition is too weak compared to the LED and thus the L2A ratio becomes too high. To human eyes, such a high L2A ratio would cause serious flare. To the camera, such a high L2A ratio causes unacceptable blooming interference which makes the On and Off LED completely indistinguishable. As mentioned earlier, such a high L2A must be avoided in VOCC for the sake of driving safety and thus needs not to be further examined in this paper.

As the L2A ratio increases, the increase of the extinction ratio and interference results in the increase of both Bvar and Wvar. This suggests that there is an optimal L2A ratio where the B2W ratio is at maximum as shown in Figure 9. This is the idea behind the optimization of VOCC systems in this paper.

#### 3.4.2. Formulation of the L2A Ratio Optimization Problem

The L2A ratio optimization in this paper is a constraint optimization problem in which the variable to be optimized is the L2A ratio, the objective function to be maximized is the B2W variance ratio, and the soft constraint is the LED interval. More specifically, the L2A ratio optimization is formulated as the problem of maximizing the B2W variance ratio with respect to the L2A ratio, subject to a given LED interval:(17)$$\begin{array}{c} \max_{r_{L2A}}\;\;\;\;\;\;\;\;\;\;\;\;\;\;R_{B2W}\left(r_{L2A}\right)\\ \mathrm{subject}\;\mathrm{to}\;\;\;\;{\mathrm{dens}}_{LED}=0.5,\ldots ,3\\ \;\;\;\;\;\;\;\;\;\;\;\;\;\;\;\;\;\;\;\;\;\;\;\;\;\;\;0.2\leq r_{L2A}\leq 4\; \end{array}$$
where $r_{L2A}$ is the L2A ratio defined as:(18)$$r_{L2A}=\frac{L_{v}}{E_{v}R},$$
$R_{B2W}$ is the B2W variance ratio defined as:(19)$$R_{B2W}=\frac{Bvar}{Wvar},$$
and $intv_{LED}$ is the normalized LED interval defined as:(20)$$intv_{LED}=\frac{d_{L2L}}{d_{LED}},$$
where $d_{L2L}$ is the distance between LEDs in the panel and $d_{LED}$ is the diameter of each LED.

In practice, a VOCC system operates in an outdoor environment having a wide range of illuminance. Given a specific LED interval $intv_{LED}$ of a particular system, based on the optimal L2A ratio ${\bar{r}}_{L2A}\left(intv_{LED}\right)$ corresponding to that LED interval, the luminance of the LED is adjusted to the optimal level ${\bar{L}}_{v}$ to obtain the lowest BER of the VOCC system:(21)$${\bar{L}}_{v}={\bar{r}}_{L2A}\left(intv_{LED}\right)E_{v}R.$$

## 4. Simulation

### 4.1. Simulation Procedure

The simulations were conducted using MATLAB version 2017b. Figure 10 describes the entire simulation procedure. In the optimization phase, a set of 100,000 LED panel sample frames was generated. These frames were replicated with randomly varying values of LED luminance, ambient light illuminance, and LED density, within given ranges. Subsequently, the frames were processed to identify the LED features, and the Bvar and Wvar were calculated. The optimal L2A ratios, subjected to different levels of LED density, were then obtained. In the testing phase, two sets of LED panel frames were generated. The first set contains frames replicated with random LED luminance values. The second set contained frames replicated with optimal LED luminance values corresponding to the given ambient light illuminance and LED density. Both sets of frames were then processed to identify the LED features and detect the transmitted bits. The resulting BER from either set was calculated to verify the effectiveness of the result of the optimization process.

### 4.2. LED Panel Image Replication

A highly realistic LED panel image replication method, shown in Figure 11, was applied in this study to provide fair analysis of the effect of the system parameters on the variances of LED features. The first step was to replicate the pixel coordinates of LEDs in the image. A pinhole camera model [12] shown in Figure 11a was used for this. The pixel coordinates of all LEDs in the image were obtained using the camera intrinsic and extrinsic parameters such as lens focal length, position, and pose, as well as the real world coordinates of the LEDs. The next step was to determine the greyscale of LED pixels in the image as shown in Figure 11b. The luminous exposure of the On and Off LED pixels and the image background pixels were determined from the LED and ambient light parameters and camera exposure settings. The Gaussian filter was then applied to replicate the blurriness phenomenon. Finally, gamma encoding was applied to calculate the greyscale of every pixel.

### 4.3. Simulation Environment

The simulation parameters are given in Table 1. The camera is assumed to have a 1 inch (13.2×8.8 mm) sensor with Full-HD (1920×1080 pixels) video recording resolution. The 18 mm lens focal length and 100 fps camera frame rate were assumed. The camera exposure time was assumed to be shorter than 1/1000 s to avoid a motion blur effect. A circular LED with a 3 cm diameter was assumed. The normalized LED interval was assumed to range from 0.5 to 2. It is important to note that the size of each individual LED is always fixed. Therefore, when the LED interval increases, the size of the entire LED panel would increases all together. The LED panel consist of 8×8 LEDs, and the LEDs were assumed to have dimming capability with their luminance ranging from 50 to 8000 cd/m${}^{2}$. The LED surface reflectance ranges from 0.3 to 0.5. The ambient light illuminance was assumed to range from 50 to 16,000 lux, which corresponds to the changing lighting conditions on an illuminated street from night to a bright sunny day. The reflectance of the background was assumed to be 0.18. This is the average value of the reflectance of middle-grey objects. The testing communication distance ranged from 30 to 70 m. The average level of blooming and blurriness of the replicated images was assumed to be 1.5 pixels. In the simulation, clear weather conditions were assumed. The transmission coefficient at 70 m was assumed to be 0.9932. As the net data rate is calculated as the product of the number of transmitted bits per frame and the camera’s frame rate, a fixed net data rate of 6.4 kbps was assumed.

Because of the wide range of ambient light illuminance and LED luminance, the L2A ratio might change greatly. However, as mentioned earlier, for the sake of driving safety, the LED luminance was adjusted to guarantee the L2A ratio was always less than 4.0.

ISO setting of the camera in the simulation is assumed to adjust automatically to ambient light. Firstly, the level of ambient light was measured using an in-camera spotlight meter. The camera exposure parameters required for yielding a middle-grey background in the image were then determined using the measured ambient light level. These camera exposure parameters include ISO speed *S*, exposure time *t*, and lens aperture *N*. Because changing lens aperture leads to change in the image’s depth of field, the lens aperture was fixed to maintain consistency in the performance of the VOCC system. The exposure time and ISO speed were increased to register more light in a dim environment, and were decreased in bright light. The range of the ISO was from 100 to 32,000. In the VOCC system, motion blur caused by the mobility of vehicles are usually not a big problem since vehicles moves in nearly the same direction. To further eliminate the motion blur problem, the longest exposure time was set to 1/1000 s.

K-means clustering was used in the simulation to detect the transmitted bits. The BERs of the system with and without L2A ratio optimization were calculated, according to the different values of the system parameters, to evaluate the effectiveness of the optimization.

In the simulation, the LED panels are assumed to be parallel to the camera. In practice, the non-parallel position between the LED panel and the camera might have negative impact on bit detection. The optimal L2A ratio for a specific non-parallel position might be slightly different to the that of the parallel case. However, vehicles are moving in parallel or near-parallel directions during most of their time on the road. The non-parallel situation only occurs in very short moments when the vehicle changes to other lane or turns to right or left. It is not reasonable for the vehicle to change the optimal L2A ratio during these very short moments and then change back to the old optimal L2A ratio when the vehicle become parallel with other vehicles again. Furthermore, in the moment that the vehicle is changing lane, the vehicle can make various angle with other receiving vehicles. As the optimal L2A ratio for different angle might be slightly different, any angle to which the vehicle chooses to change the optimal L2A ratio will not bright the optimal result to receiving vehicles and other angles. Consequently, in practice, the vehicle should only stick with the optimal L2A ratio for the parallel position.

### 4.4. Simulation Results

#### 4.4.1. Effect of L2A Ratio

This section provides simulation results to illustrate the effect of the L2A ratio. The LED panel images were replicated with the L2A ratio varying from 0.2 to 4. The ambient light is set randomly in the range from 50 to 16,000 lux. The LED luminance is set based on the ambient light and the L2A ratio. The LED surface reflectance is set randomly in the range from 0.3 to 0.5. The ISO is set based on the ambient light. The value of the communication distance and LED interval are fixed at 50 m and 1, respectively. The remaining parameters are set to their fixed values given in Table 1. Figure 12 shows examples of the LED panel frames replicated with L2A ratios of 0.2, 1, 2, and 4, respectively. The figure shows that the extinction ratio changes according to the L2A ratios: At a low L2A ratio, the extinction becomes low because the greyscales of the On and Off LEDs are almost the same. On the other hand, at a high L2A ratio, the extinction becomes high, expressed through the clear difference in the greyscales of the On and Off LEDs.

Figure 13 shows examples of the distribution of the LED features of typical frames at different L2A ratios. The LED interval in these figures are about the same. In Figure 13a, the L2A ratio is small and the features of each class of LEDs clump together, leading to a small Wvar. However, the clusters of the On and Off LEDs are also close to each other, leading to a small Bvar. In Figure 13b, the L2A ratio is at a mid level and the features of each class of LEDs are spread more, leading to a larger Wvar. However, the clusters of the On and Off LEDs are also apart from each other, leading to a larger Bvar. In Figure 13c, the L2A ratio is high, the clusters of each class of LEDs are spread even more, leading to an even larger Wvar. The Bvar also increases with the distance between the centers of the two clusters.

Figure 14a shows trend of Bvar and Wvar when the L2A ratio increases. The Wvar of the LED features can be seen to increase all the way as the L2A ratio increases. The Bvar also initially increases with the L2A ratio. However, when the L2A ratio exceeds 2.5, the Bvar barely increases. This is because the maximum greyscale in the image is 255. When the L2A ratio exceeds 2.5, the greyscales of the On LEDs do not increase anymore. However, due to increasing interference, the greyscales of the Off LEDs continue increasing and approach those of the On LEDs. This makes the clusters of the Off LED features become similar to those of the On LED features and reduces the Bvar.

Figure 14b shows the trend of B2W variance ratios when the L2A ratio changes. It can be seen that when the LED interval is 1 and the communication distance is 50m, the value of the optimal L2A ratio is 2. The distinguishability of the On and Off LEDs would be highest at this L2A ratios. Note that this optimal value of the L2A ratio would change when the LED interval and the communication distance change.

#### 4.4.2. Effect of LED Interval and Communication Distance

##### (a) Effect of LED Interval

This section provides simulation results to illustrate the effect of LED interval. The LED panel images were replicated with LED interval varying from 0.5 to 2. The communication distance and the L2A ratio are fixed at 50m and 1, respectively. The ambient light is set randomly in the range from 50 to 16,000 lux. The LED luminance is set based on the ambient light and the L2A ratio. The LED surface reflectance is set randomly in the range from 0.3 to 0.5. The ISO is set based on the ambient light. The remaining parameters are set to their fixed values given in Table 1.

Figure 15 shows examples of LED panel frames replicated with LED intervals of 0.5, 1, 1.5, and 2, respectively. Note that the size of each individual LED is always fixed. Therefore, the size of the entire LED panel would increases as the LED interval increases. For the sake of clear illustration, the image of LED panels with smaller intervals, which have smaller sizes, are zoomed closer to keep the size of all LED panel images the same as shown in Figure 15. Because of the differences in the degree of zooming, images of LED panels in the left, which has smaller intervals, appear blurrier compared to that in the right in Figure 15. From this figure, it is obvious that the LED panels with smaller interval would have less LED distinguishability.

Figure 16 shows examples of the distributions of LED features in ten frames replicated with different LED intervals but the same L2A ratio of 1 and the same communication distance of 50 m.

Figure 17a shows the Bvar and Wvar at these LED intervals. The clusters of the On and Off LED features can be seen to become more compact as the interval increases. Therefore, the Wvar would decrease greatly as the interval increases. As the clusters of the On and Off LED features become more compact, the distance between the centers of the two clusters increases as the interval increases. However, an interesting phenomenon can be observed in Figure 16d: At the LED interval of 2, the clusters of the LED features appear to become more compact and the distance between the two clusters decreases. This is caused by two phenomena: The range of the GRI (which measures the interference between LEDs) decreases due to the decreasing interference. The range of the AGR (which measures the average greyscale in each LED region) also decreases. Recall from Figure 2, the entire LED panel is divided into equal rectangular LED regions. As the LED interval increases, these rectangular regions become larger and decrease the average greyscale in each region. The decrease in the range of GRI and AGR leads to the decrease in the distance between the On and Off LED clusters, as well as the decrease in Bvar, as shown in Figure 17a. However, this would not negatively impact the separation of the On and Off LEDs as Figure 17b shows.

Figure 17b shows the B2W variance ratio at different LED intervals. Because Bvar increases and Wvar decreases, the B2W variance ratio greatly decreases as the interval increases. At the LED interval of 2, Bvar decreases. However, the great decrease of Wvar causes the B2W variance ratio to eventually increase sharply.

##### (b) Effect of Communication Distance

This section provides simulation results to illustrate the effect of communication distance. The LED panel images were replicated with the communication distance varying from 30 to 70 m. The L2A ratio of 1 and the LED interval of 1 are assumed in all replicated frames. The ambient light is set randomly in the range from 50 to 16,000 lux. The LED luminance is set based on the ambient light and the L2A ratio. The LED surface reflectance is set randomly in the range from 0.3 to 0.5. The ISO is set based on the ambient light. The remaining parameters are set to their fixed values given in Table 1.

Figure 18 shows examples of LED panel frames replicated at communication distances of 30, 50, and 70 m, respectively. As both LED interval and communication distance affect the LED feature variance through LED spacing, the effect of these two parameters are similar. More specifically, increasing the distance has the same effect as decreasing LED interval. When the distance increases, the LED panel image becomes smaller and thus makes the LED spacing smaller. Consequently, the LED interference gets stronger as the distance increases.

Figure 19 shows examples of the distribution of LED features in ten frames replicated with different communication distance but the same L2A ratio of 1 and the same LED interval of 1. Because of the increase of the LED interference as the distance increases, the cluster of each class of LED spread spreads more, which makes the Wvar increases. Moreover, because of these spreading, the clusters of On and Off LED features get closer to each other, making the Bvar decreases.

Figure 20a shown the Bvar and Wvar at different communication distances. As suggested from Figure 19, the Bvar decreases while the Wvar increases as the distance increases. This makes the B2W variance ratio decreases sharply as the distance increases as shown in Figure 20b.

#### 4.4.3. Optimal L2A Ratio at Different LED Intervals

This section provides simulation results to illustrate the effect of distance and LED interval on the optimal L2A ratio. The LED panel images were replicated with the L2A ratio varying from 0.2 to 4, communication distance varying from 30 to 70 m, LED interval varying from 0.5 to 2. The ambient light is set randomly in the range from 50 to 16,000 lux. The LED luminance is set based on the ambient light and the L2A ratio. The LED surface reflectance is set randomly in the range from 0.3 to 0.5. The ISO is set based on the ambient light. The remaining parameters are set to their fixed values given in Table 1.

Since the Bvar and Wvar are affected by the LED interval and communication distance, the optimal L2A ratio is subject to change when these two parameters change. Figure 21a shows the optimal L2A at different LED intervals and communication distances. It can be seen that the optimal value of the L2A ratio tends to be higher at short communication distances and large LED intervals. This is because at these conditions, the LED interference is low and thus the positive impact of the L2A ratio on LED extinction ratio is only outweighed by its negative impact on LED interference at a high L2A ratio.

Since the optimal L2A ratio varies across different distances as shown in Figure 21a, each LED interval has a range of the optimal L2A ratio as shown in Figure 21b. For each LED interval, the average value in the corresponding range is determined as the optimal L2A ratio to be applied for a running VOCC system.

#### 4.4.4. BER at Different Communication Distances

The effectiveness of optimizing the L2A ratio was verified through examining the BER of a VOCC system. In this simulation, the communication distance ranged from 30 to 70 m. At each communication distance, two sets of LED frames were replicated. In the optimal set, the LED luminance was adjusted to obtain the average optimal L2A ratio given in Figure 21b. In the non-optimal set, the LED luminance was adjusted to a random L2A ratio in the range from 0.2 to 4. In both sets, the LED interval is set randomly from 0.5 to 2. The ambient light is set randomly in the range from 50 to 16,000 lux. The LED luminance is set based on the ambient light and the L2A ratio. The LED surface reflectance is set randomly in the range from 0.3 to 0.5. The ISO is set based on the ambient light. The remaining parameters are set to their fixed values given in Table 1.

The results are shown in Figure 22: The BER of the optimal set is seen to be lower than that of the non-optimal set at all communication distances. Because of L2A ratio optimization, this improvement in BER comes without the costs of reducing the data rate or increasing the size of the LED panel on the vehicle.

## 5. Conclusions

In a VOCC system, the digital bits 1 and 0 are represented by the On and Off status of LEDs. The On and Off statuses of LEDs in a received image need to be determined in order to obtained the bits transmitted from one vehicle to another. Lowering the BER is obviously the goal of all VOCC system designers. One might intuitively assume that increasing the LED power would make the On and Off LEDs more distinguishable and thus reduce the BER. However, this paper shows that it is not that simple for VOCC. Increasing the LED luminance is known to reduce the BER because the LED extinction ratio increases. However, due to the nature of image acquisition, a higher LED luminance also leads to higher LED interference, which has negative impact on BER. Consequently, an optimal ratio of LED to ambient light (L2A) exists in which the BER is at its lowest. Through theoretical analysis, this paper elucidates that besides the L2A ratio, LED interval is another parameter with a decisive effect on BER. Furthermore, this paper reveals that the lowest BER can be achieved through maximizing the between-class to within-class (B2W) variance ratios of the On and Off LED features in received images. Based on these analyses, we formulated a constrained optimization method of finding the optimal L2A ratio to maximize the B2W variance ratio subject to different LED intervals. Through simulations, it is verified that by applying these optimal L2A ratios, much lower BERs of a VOCC system can be achieved without changing the data rate.

VOCC system designers might wonder if the optimal L2A ratios reported in this paper are what should be directly applied to their system. The answer is no. While the optimal L2A ratios suggested in this paper are likely to provide better performance than a random one, they might not necessarily be the optimal ratios for their system. Even with the exact LED intervals in this paper, the LED interference might be slightly different in their system due to different LED coatings, different LED panels, different auto-ISO mechanisms of cameras, or other reasons. However, it is worth noting that the most important contributions of this paper are not the optimization results themselves but the whole idea of the necessity of L2A optimization, the theoretical analysis of the effect of the L2A ratio on bit detection accuracy, and the means to obtain the optimal L2A ratio in the VOCC system.

To the best of our knowledge, this paper is the first work in the literature to discuss the necessity of LED dimming, the proper auto exposure mechanism of a camera, and the effect of L2A ratios in VOCC. Due to many fundamental difficulties, especially the wireless synchronization between multiple cameras and LEDs, studies on VOCC are still focusing on the simplest version of the system, which involves the communication between two vehicles with one transmitting the LED panel and one receiving the camera. In such a simple system, the LED luminance can be fixed, the camera exposure can be manually adjusted to the LED luminance and thus the L2A ratio problem can be neglected. In a real VOCC system where the communication happens between multiple vehicles with various cameras and LED panels, the camera exposure and the LED luminance need to be adjusted to the ambient light and the L2A ratio will become a deciding factor for the system performance. This paper not only introduces the idea of L2A ratio optimization to the research community but also provides mathematical analysis to elucidate the effect of the L2A ratio on the bit detection accuracy of the VOCC system. Through this paper, VOCC system designers would know that instead of trying to use more powerful LEDs, they need to find the optimal L2A ratio to achieve the lowest BER. This task can be achieved by using the proposed optimization method through maximizing the B2W variance ratio as described in the paper.

## Figures and Tables

**Figure 1 sensors-20-00292-f001:**
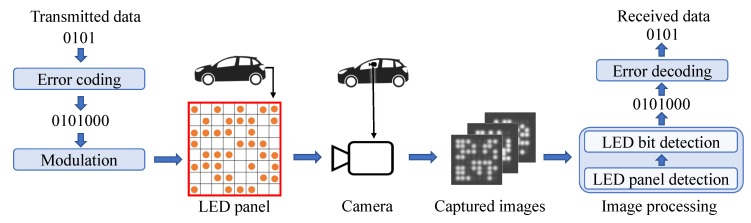
VOCC system architecture.

**Figure 2 sensors-20-00292-f002:**
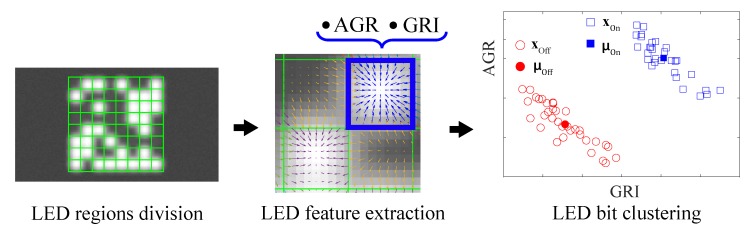
LED bit detection algorithm.

**Figure 3 sensors-20-00292-f003:**
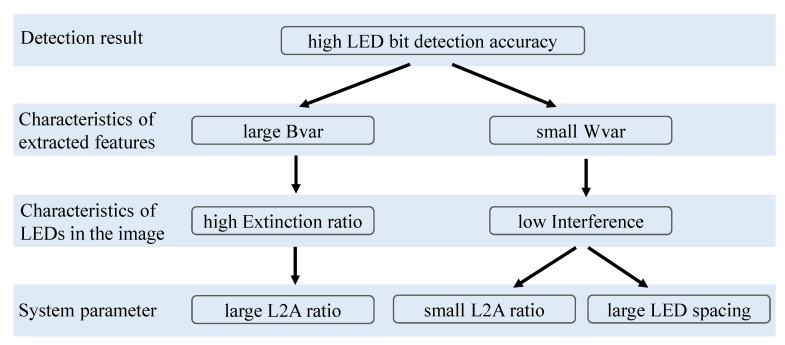
Decisive factors of LED bit detection accuracy.

**Figure 4 sensors-20-00292-f004:**
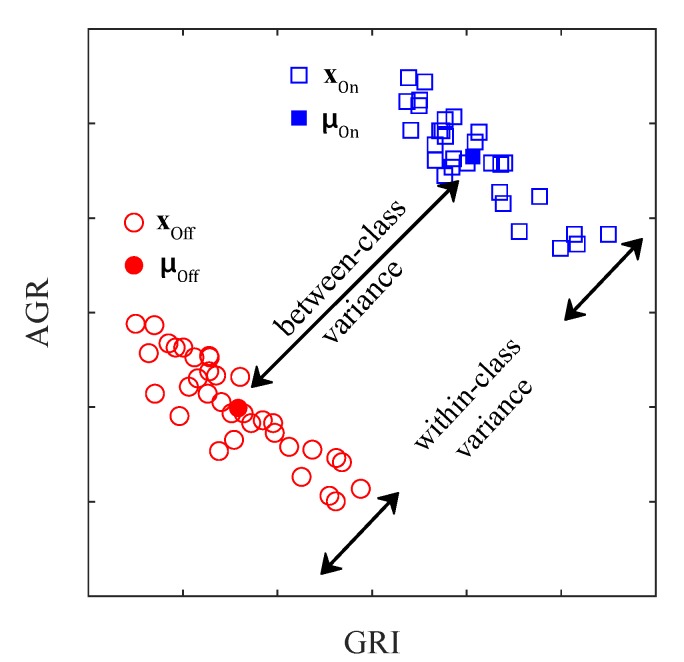
Between-class variance and within-class variance.

**Figure 5 sensors-20-00292-f005:**
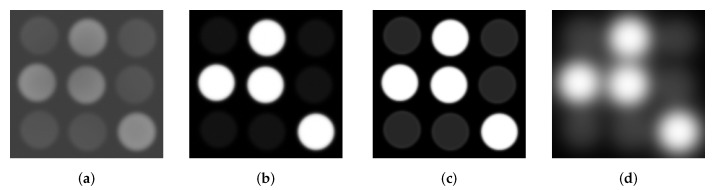
Extinction ratio and interference of LED panel images. (**a**) Low extinction ratio. (**b**) High extinction ratio. (**c**) Low interference. (**d**) High interference.

**Figure 6 sensors-20-00292-f006:**
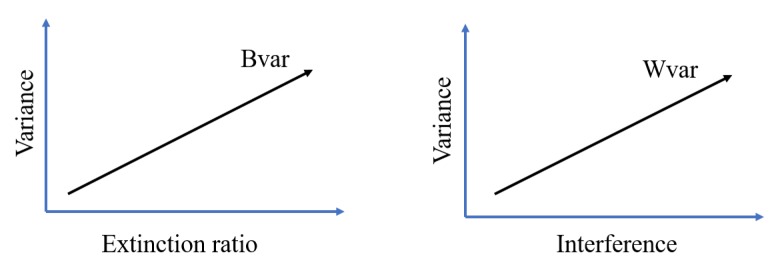
Effect of extinction ratio and interference on between-class variance (Bvar) and within-class variance (Wvar).

**Figure 7 sensors-20-00292-f007:**
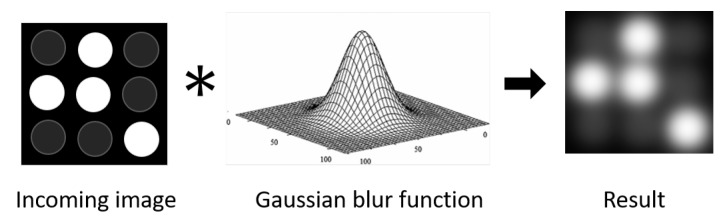
Image interference.

**Figure 8 sensors-20-00292-f008:**
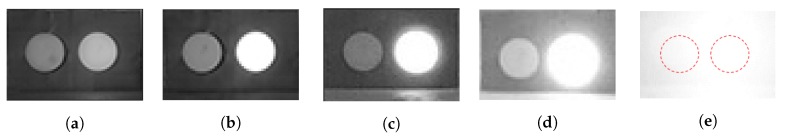
Contradictory effect of the light to ambient light (L2A) ratio on extinction ratio interference. (**a**) Direct strong sunlight. (**b**) Direct mild sunlight. (**c**) Indirect sunlight. (**d**) Overcast day. (**e**) Twilight.

**Figure 9 sensors-20-00292-f009:**
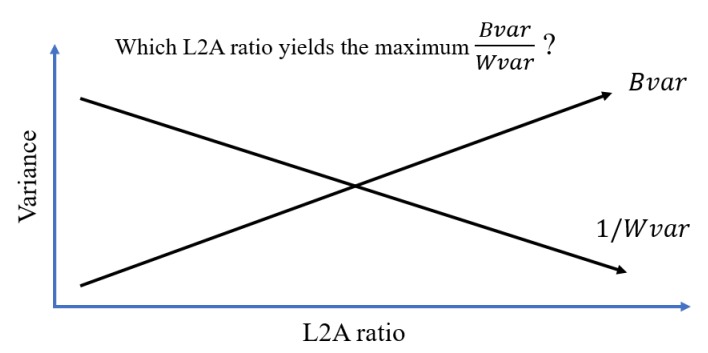
The idea behind vehicular optical camera communication (VOCC) system optimization.

**Figure 10 sensors-20-00292-f010:**
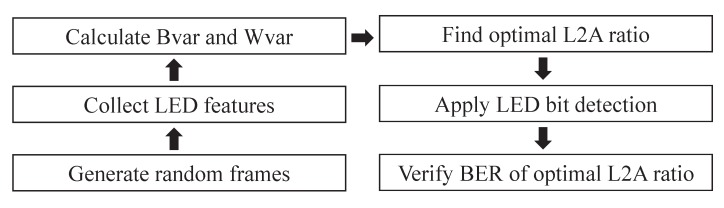
Simulation procedure.

**Figure 11 sensors-20-00292-f011:**
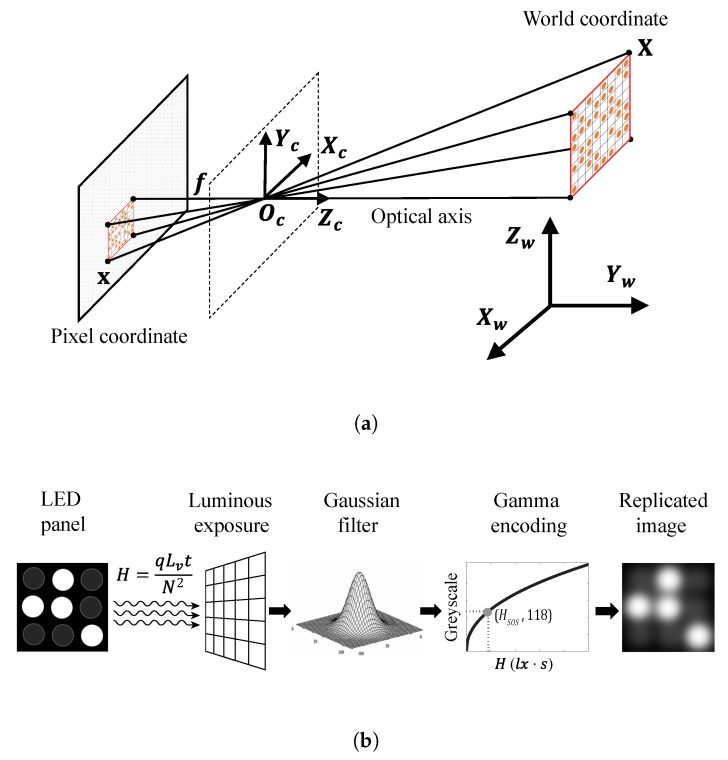
LED panel image replication. (**a**) Pinhole camera model to determine LED pixel coordinates. (**b**) LED blooming interference replication and gamma encoding.

**Figure 12 sensors-20-00292-f012:**
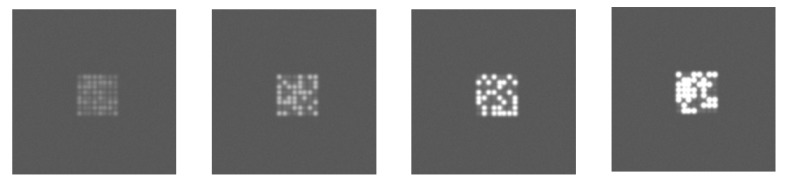
Replicated LED panel frames at different L2A ratios.

**Figure 13 sensors-20-00292-f013:**
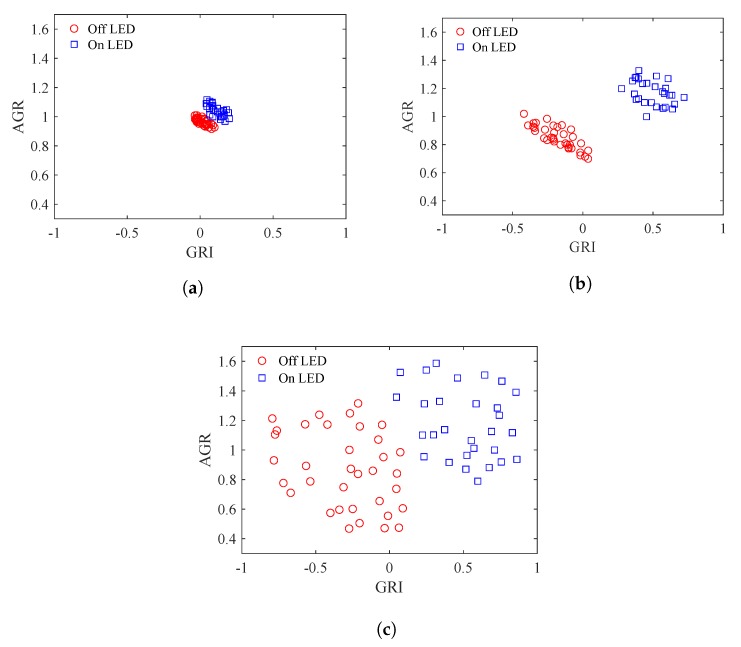
Distribution of LED features at different L2A ratios. (**a**) Low L2A ratio. (**b**) Mid L2A ratio. (**c**) High L2A ratio.

**Figure 14 sensors-20-00292-f014:**
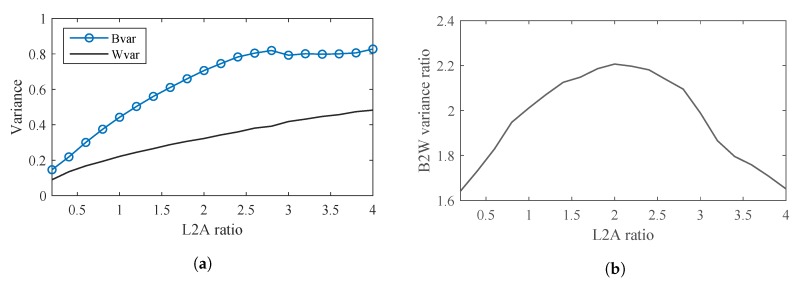
Variances corresponding to L2A ratios. (**a**) Bvar and Wvar corresponding to L2A ratios. (**b**) Between-class variance to the within-class variance (B2W) ratio corresponding to L2A ratios.

**Figure 15 sensors-20-00292-f015:**

LED panel frames at different LED intervals.

**Figure 16 sensors-20-00292-f016:**
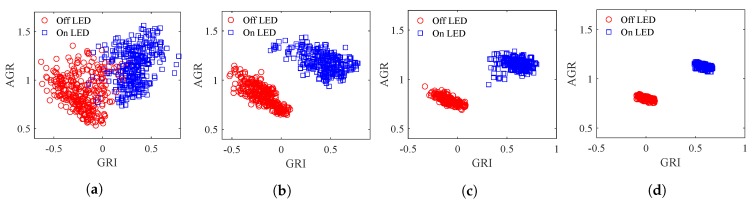
Distribution of LED features at different LED intervals. (**a**) LED interval = 0.5. (**b**) LED interval = 1. (**c**) LED interval = 1.5. (**d**) LED interval = 2.

**Figure 17 sensors-20-00292-f017:**
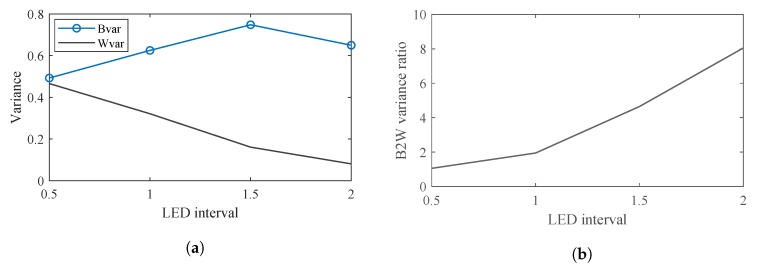
Variances corresponding to LED intervals. (**a**) Bvar and Wvar corresponding to LED intervals. (**b**) B2W variance ratio corresponding to LED intervals.

**Figure 18 sensors-20-00292-f018:**

LED panel frames at different distances.

**Figure 19 sensors-20-00292-f019:**
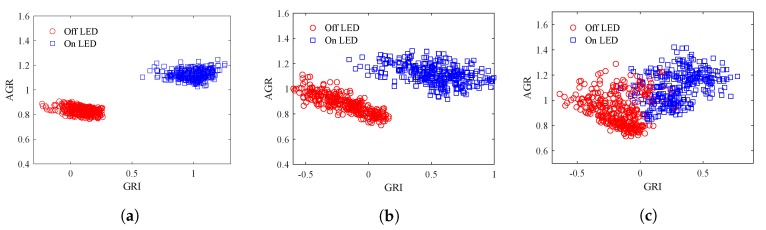
Distribution of LED features at different communication distances. (**a**) Distance = 30 m. (**b**) Distance = 50 m. (**c**) Distance = 70 m.

**Figure 20 sensors-20-00292-f020:**
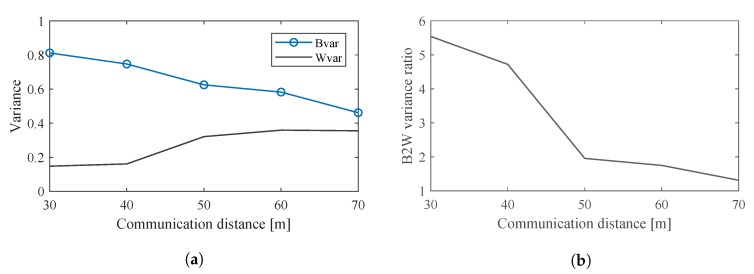
Variances at different distances. (**a**) Bvar and Wvar at different distances. (**b**) B2W variance ratio at different distances.

**Figure 21 sensors-20-00292-f021:**
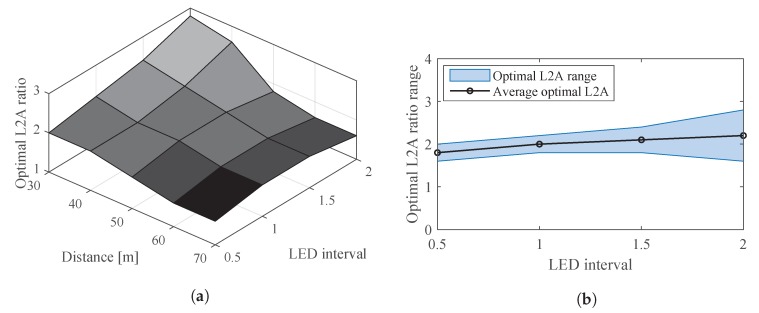
Optimal L2A ratio. (**a**) Optimal L2A ratio at different LED intervals and distances. (**b**) Optimal range of L2A ratio corresponding to LED intervals.

**Figure 22 sensors-20-00292-f022:**
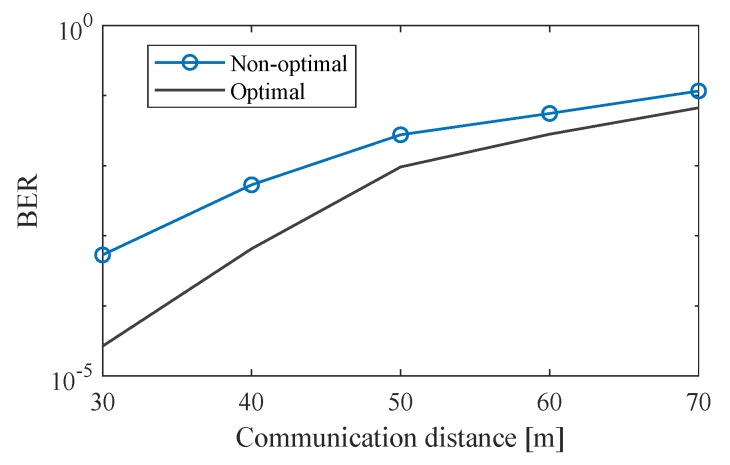
BER at different communication distances.

**Table 1 sensors-20-00292-t001:** Simulation environment.

Parameters	Values
Sensor physical size	13.2×8.8 mm
Sensor resolution	1920×1080 (pixels)
Lens focal length	25 mm
Frames per second	100 fps
Longest exposure time	1/1000 (s)
ISO setting	Auto from 100 to 3200
Gamma encoding value	2.2
LED shape	circle
LED diameter	3 (cm)
LED interval	0.5 to 2
Number of LEDs in the array	8×8 (LEDs)
LED luminance	50 to 8000 ($\mathrm{cd}/m^{2}$)
LED surface reflectance	0.3 to 0.5
Ambient light illuminance	50 to 16000 (lux)
Average background reflectance	0.18
Communication distance	30 to 70 (m)
Blooming level	1.5
Net data rate	6.4 (kbps)
Transmission coefficient at 70 m	0.9932
Range of L2A ratio	0.2 to 4

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
