# Peer review of "The Necessity of LED to Ambient Light Ratio Optimization for Vehicular Optical Camera Communication"

_sensors, 2020, doi:10.3390/s20010292_

Round 1
Reviewer 1 Report
Summary:
The paper studies the optimization criterion for robust LED detection and signal demodulation in vehicular optical camera communication. The paper studies the tradeoff between LED to ambient light intensity modulations and presents simulation studies of different cases where signal detection is more amenable than others.
Strengths:
The paper explores a credible problem which is timely. The study of LED to ambient light intensity can be helpful to the community in advancing vehicular camera communication. The paper is well organized.
Weaknesses:
The work is largely based on a hypothesis which hasn't been tested. The dataset is largely limited to controlled settings from which simulation experiments have been derived. It would have been better to conduct empirical measurements to see how well they can be characterized. The results are not surprising and are as expected.
Detailed Comments:
The work is based on a hypothesis that there is essentially a tradeoff in signal demodulation quality between LED and ambient light intensities. The authors choose the statistical and ratio metrics using these intensities without any proof or supportive arguments as to why these are being chosen. The parameters such as extinction ratio have not been defined clearly. The usage is vague across the paper and creates a confusion. It will be helpful to have a table listing all parameters and metrics used and their corresponding definitions if they warrant one. It is unclear why the authors choose to run a simulation study instead of capturing images using a camera and across different LED light intensity modulations. The simulation results assume very controlled distortions which are primarily modeled as Gaussian blur in the paper. It is for this reason that an empirical evaluation would help capture the real world distortions in such settings. What happens to the insights and findings during mobility has not been discussed.
Author Response
The authors would like to thank the editor and reviewers for valuable comments to help the authors to improve the manuscript. The concerns of the reviewer are addressed in the attached file.

Reviewer 2 Report
This main topic of this paper is concerned with optimizing VOCC system parameters to obtain the optimal BER without changing or increasing the system implementation costs.It claims that the L2A ratio and LED intervals are two key parameters which have decisive effects on BER. The paper analysis detailedely how the L2A ratio and LED interval influence the LED distinguishability and formulate the issue as a constrained optimization problem.
Although the presented system model was built in a simple and straight way (many practical factors in the VOCC system are not considered), the analysis of how each parameters affects the commnication performance are detailedly discussed and validated with adequate simulation experiments.
This paper is well organized with a very good presentation. The goal of this paper is very clear, and the analysis and simulation answer the addressed problem in a consistent way. This paper could be reconsider for publication after the following concerns have been well answered:
1: As addressed in the paper, the L2A ratio is a very important factor that affect the BER. However, the "ambient light" is a very general phrase, in fact, it is a very dynamic variable in any practial VOOC system and is difficult to quantify. What would be to price of obtaining the optimal L2A ratio? How about the responsivity with regards to the communication rate?
2: Have you consider the case when the LED panel and the camera are not in a parallel position? What about the image distortion? I think these are very common problem in real applications. What would be the effect on the optical L2A and the final BER?
3: I suggest that the authors could set-up some real experimentation tests to further validate and prove the obtaining results. VOOC system is a very practical system, the overall performance of the system (BER) can be affected by various factors, like weather, temperature, vibration, tilt,image distortion, etc.al. A real demonstration experiment would be more convincing.
4: Figure 11: It is better to detail on the picture what's the L2A ratio of each figure. The same of figure 14 and 17.
5:For each of the simulation (Section 4.4.1-4.4.4), it is highly recommended to detail the values of each parameter for obtaining the presented results. It is not enough by just saying "The other system parameters, including ambient light, LED surface reflectance, and LED interval were set randomly within their ranges given in Table I."
Author Response

(The authors gave the same response as above.)

Round 2
Reviewer 1 Report
The response to reviewer comments are mostly satisfactory. The changes made are helpful to the reader. However, broadly, considering the scope of this work and the results, it is still unclear why these simulation results are important. Authors themselves agree in their response that the results are more or less as one would expect. If there is nothing surprising about the result how does this help educate the community. Moreover, the lack of experimental evidence demerits the outcome of this paper significantly. While the short revision time is understood, the authors could have conducted a simple experiment that shows the variation in the extinction ratio across indoor and outdoor ambient lighting conditions.
I am not convinced that this work will advance the VLC knowledge base as it seems to discuss what may be known -- perhaps thats why prior work did not explicitly discuss the L2A relationships.
Author Response
The authors would like to thank the reviewer for valuable comments to help the authors to improve the manuscript. We have addressed the reviewer's concern in the attached file.
